# Saliva Is a Valid Alternative to Nasopharyngeal Swab in Chemiluminescence-Based Assay for Detection of SARS-CoV-2 Antigen

**DOI:** 10.3390/jcm10071471

**Published:** 2021-04-02

**Authors:** Alessandra Amendola, Giuseppe Sberna, Eleonora Lalle, Francesca Colavita, Concetta Castilletti, Giulia Menchinelli, Brunella Posteraro, Maurizio Sanguinetti, Giuseppe Ippolito, Licia Bordi, Maria Rosaria Capobianchi

**Affiliations:** 1Laboratorio di Virologia, Istituto Nazionale per le Malattie Infettive (INMI) Lazzaro Spallanzani IRCCS, 00149 Rome, Italy; alessandra.amendola@inmi.it (A.A.); giuseppe.sberna@inmi.it (G.S.); eleonora.lalle@inmi.it (E.L.); francesca.colavita@inmi.it (F.C.); concetta.castilletti@inmi.it (C.C.); maria.capobianchi@inmi.it (M.R.C.); 2Dipartimento di Scienze di Laboratorio e Infettivologiche, Fondazione Policlinico Universitario A. Gemelli IRCCS, 00168 Rome, Italy; giulia.menchinelli@hotmail.it (G.M.); maurizio.sanguinetti@unicatt.it (M.S.); 3Dipartimento di Scienze Biotecnologiche di Base, Cliniche Intensivologiche e Perioperatorie, Università Cattolica del Sacro Cuore, 00168 Rome, Italy; Brunella.Posteraro@Unicatt.it; 4Dipartimento di Scienze Mediche e Chirurgiche, Fondazione Policlinico Universitario A. Gemelli IRCCS, 00168 Rome, Italy; 5Direzione Scientifica, Istituto Nazionale per le Malattie Infettive (INMI) Lazzaro Spallanzani IRCCS, 00149 Rome, Italy; giuseppe.ippolito@inmi.it

**Keywords:** SARS-CoV-2, antigen detection, RT-PCR, saliva samples, diagnosis

## Abstract

Diagnostic methods based on SARS-CoV-2 antigens detection are a promising alternative to SARS-CoV-2 RNA amplification. We evaluated the automated chemiluminescence-based Lumipulse^®^ G SARS-CoV-2 Ag assay on saliva samples, using Simplexa™ COVID-19 Direct assay as a reference test. Analytical performance was established on a pool of healthy donors’ saliva samples spiked with the 2019-nCoV/Italy-INMI1 isolate, whereas clinical performance was assessed on fresh saliva specimens collected from hospitalized patients with suspect or confirmed COVID-19 diagnosis. The limit of detection (LOD) was 0.65 Log TCID50/mL, corresponding to 18,197 copies/mL of SARS-CoV-2 RNA. Antigen concentrations and SARS-CoV-2 RNA were highly correlated (r = 0.99; *p* < 0.0001). Substantial agreement (80.3%) and significant correlation (r = −0.675; *p* = 0.0006) were observed between Lumipulse^®^ G assay results and Ct values on clinical samples, with 52.4% sensitivity and specificity 94.1%. Sensitivity exceeded 90.0% when calculated on samples with Ct < 25, and specificity was 100% when excluding samples from recovered patients with previous COVID-19 diagnosis. Overall, chemiluminescence-based antigen assay may be reliably applied to saliva samples to identify individuals with high viral loads, more likely to transmit the virus. However, the low positive predictive value in a context of low SARS-CoV-2 prevalence underscores the need for confirmatory testing in SARS-CoV-2 antigen-positive cases.

## 1. Introduction

As the ongoing COVID-19 pandemic is growing fast, accounting for more than 121 million laboratory-confirmed cases and more than 2.69 million deaths reported around the world (https://www.worldometers.info/coronavirus, accessed on 18 March 2021), reliable and rapid detection methods are increasingly necessary to diagnose and track patients with COVID-19 worldwide. Although WHO guidelines recommend using SARS-CoV-2 RNA amplification tests for diagnostic purposes [1], molecular methods are relatively expensive and time-consuming and need expert personnel with specialized equipment, thus prompting the development of point-of-care (POC) testing methods [2]. Among these, rapid antigen detection tests currently deserve great attention because they are intrinsically less laborious, require a few minutes to results and have the potential to satisfy the pressing demand for early SARS-CoV-2 infection diagnosis [3,4,5]. It is worthy to note that, in some countries, rapid antigen detection tests are suggested as first-line diagnostic testing [6], and recently ECDC recommended clinical validations of rapid antigen tests (https://www.ecdc.europa.eu/en/publications-data/options-use-rapid-antigen-tests-covid-19-eueea-and-uk, accessed on 19 November 2020).

In SARS-CoV-2 diagnosis, saliva has entered the shortlist of clinical samples to which apply the current laboratory tests since recent studies have shown that molecular tests performed on saliva had sensitivity and specificity comparable to those observed with nasopharyngeal swab samples [7,8,9,10,11,12]. Therefore, on 8 May 2020, the U.S. Food and Drug Administration authorized the first diagnostic molecular test for COVID-19 testing, issuing an emergency use authorization (EUA) with the option of using home-collected saliva samples (https://www.fda.gov/media/137773/download, accessed on 13 November 2020).

In Italy, following health authorities mandate, we evaluated the performance of a novel antigenic test using saliva samples, namely the Lumipulse^®^ G SARS-CoV-2 Ag assay (Fujirebio, Tokyo, Japan), which received the CE marking for qualitative and quantitative detection of the SARS-CoV-2 nucleoprotein (N) antigen on both saliva and nasopharyngeal swab samples, based on the chemiluminescent enzyme immunoassay technology. First, we established the analytical sensitivity of the Lumipulse^®^ G SARS-CoV-2 Ag assay with healthy donors’ saliva specimens spiked with SARS-CoV-2. Second, we analyzed the Lumipulse^®^ G SARS-CoV-2 Ag assay results obtained on saliva samples collected from patients hospitalized with suspected COVID-19 diagnosis. All data were then compared with those obtained with the Simplexa™ COVID-19 Direct assay (Diasorin Molecular, Saluggia, Italy), a molecular procedure previously validated in our laboratory for use on saliva [12] and currently the only molecular assay CE licensed for the use of these specimens, from our knowledge.

## 2. Materials and Methods

### 2.1. Saliva Samples

Saliva samples (either frozen or fresh) were collected from healthy donors and patients admitted to the National Institute for Infectious Diseases “L. Spallanzani” (INMI) in Rome with suspected COVID-19 infection (Table 1).

All saliva samples were anonymized prior to analysis. Samples were mostly collected via passive drooling and spontaneously produced without external stimuli; to obviate scarce salivation in some patients, sublingual oral fluid was collected using a sterile pipette. All samples were collected without the addition of any type of diluent, at least 30 min after drinking or eating or washing teeth; some of them were maintained at 4 °C for 1 to 3 days before testing (fresh saliva samples) or frozen after molecular assay and retrospectively tested for the presence of antigens (frozen saliva samples).

### 2.2. Preparation of Virus-Spiked Saliva Samples from Healthy Donors

Fresh saliva samples obtained from healthy donors spiked with different concentrations of the 2019-nCoV/Italy-INMI1 isolate [13] were used to study the analytical sensitivity of the Lumipulse^®^ G SARS-CoV-2 Ag assay. The SARS-CoV-2 isolate 2019-nCoV/Italy-INMI1 [13] was propagated in Vero E6 cells (C1008; African green monkey kidney cells). Cells were maintained in Dulbecco’s minimal essential medium (DMEM) containing 10% fetal bovine serum (FBS), 0.05 mg/mL gentamicin, at 37 °C with 5% CO2. FBS concentration was reduced to 2% for viral propagation. The infectious titer of the viral preparation propagated in Vero E6 cells was 10^7^ tissue culture 50% infective dose (TCDI_50_)/mL, according to the Reed-Muench method. To establish the viral RNA content of the viral preparation, SARS-CoV-2 RNA was extracted and amplified by quantitative real-time RT-PCR in Rotor-GeneQ Real-Time cycler (Qiagen, Hilden, Germany), using the RealStar^®^ SARS-CoV-2 RT-PCR Kit 1.0 (Altona Diagnostic GmbH, Hamburg, Germany) assay. According to a standard curve prepared through serial dilutions of the SARS-CoV-2 E gene [14] obtained by European Virus Archive-GLOBAL, the viral preparation resulted in containing 10^10^ RNA copies/mL.

Before spiking with SARS-CoV-2 particles from the aforementioned viral stock, a pooled sample of saliva was obtained from 21 healthy donors by mixing samples together, centrifuging them at 2000× *g* for 5 min, then diluting 1:1 with the diluent supplied from the Fujirebio kit. Serial ten-fold dilutions, to obtain viral concentrations from 10^6^ down to 10^−1^ TCDI50/mL, were prepared for testing in triplicates with the Lumipulse^®^ G SARS-CoV-2 Ag (see below). When established the last dilution with 100% of positive results, obtained at 10 TCID_50_/mL, at least five replicates of serial 1:2 dilutions were performed until reaching 1 TCID_50_/mL (Table 2). These results were used to calculate the limit of detection (LOD) of the assay by Probit analysis. Overall, the equivalence between N antigen concentration and RNA copies, referred to as the viral stock preparation, was 1 pg = 4.24 log copies, corresponding to 17,378 copies.

### 2.3. Lumipulse^®^ G SARS-CoV-2 Ag Assay on Saliva Samples

The Lumipulse^®^ G SARS-CoV-2 Ag assay was used according to the manufacturer’s instructions. Briefly, saliva samples were centrifuged at 2000× *g* for 5 min, and 100 μL of supernatant was diluted 1:1 with the diluent (100 μL) supplied from the Lumipulse^®^ G SARS-CoV-2 Ag assay kit and loaded into the Lumipulse^®^ G1200 automated system, able to process 120 samples per hour, providing the first result after 30 min. Following reaction with anti-SARS-CoV-2 Ag monoclonal antibody-coated magnetic particles, reaction signals allowed quantitative measurement of SARS-CoV-2 Ag in the sample. The analytical sensitivity declared by the manufacture for saliva is 0.19 pg/mL, with a cut-off set at 0.67 pg/mL; the claimed linear range was up to 6056.64 pg/mL. Results below the cut-off value were considered negative, and those above 0.67 pg/mL were considered positive.

### 2.4. Simplexa™ COVID-19 Direct Assay on Saliva Samples

In a previous article, we compared the Simplexa™ COVID-19 Direct assay with the RT-PCR method established by Corman et al. [14], demonstrating substantial concordance [κ = 0.8; 95% Confidence Interval (CI) 0.6–0.9] between the two assays, with a LOD for the Simplexa™ COVID-19 Direct assay of 1905 copies/mL viral RNA (Target S gene) [12]. Therefore, we used the Simplexa™ COVID-19 Direct assay as a reference method, also considering that it was the only commercial molecular assay with a CE-IVD mark for use on saliva in the days we were carrying our comparison analysis. This assay is a real-time RT-PCR system that enables the direct amplification of SARS-CoV-2 RNA without sample processing like RNA extraction. Two different regions of the SARS-CoV-2 genome were amplified: ORF1ab and S gene; an RNA internal control is used to detect RT-PCR failure and/or inhibition. The Simplexa™ COVID-19 Direct assay was used according to the manufacturer’s instructions. Briefly, one vial of Reaction Mix was thawed for each sample, followed by loading 50 μL of a saliva sample that was previously diluted 1:1 with 0.9% NaCl and 50 μL of Reaction Mix to their specific wells on a direct amplification disk (DAD). The DAD was then loaded onto the LIAISON^®^ MDX instrument (DiaSorin Molecular). Upon completion of the run, the software automatically calculated and provided easy-to-understand results with the ability to check amplification curves after a run. Samples with Ct values < 40 were considered positive. For statistical calculations (i.e., Mann-Whitney test), an arbitrary value of 45 Ct was assigned to all negative samples (i.e., those with Ct > 40).

### 2.5. Statistical Analysis

Data management and analyses were performed using GraphPad Prism version 8.00 (GraphPad Software, La Jolla, CA, USA). The analytical sensitivity (SARS-CoV-2 copy number and TCDI50 at a 95% detection rate) was calculated by Probit analysis, using the MedCalc statistical software (version 19.6, MedCalc Software Ltd., Ostend, Belgium). The evaluation of the qualitative concordance between results was performed using the weighted Cohen’s kappa statistics and its 95% CI; the agreement was evaluated as poor (less than 0.50), moderate (0.50–0.74), substantial (0.75–0.90), and almost perfect if greater than 0.90. Correlation analyses were performed using linear regression analysis. A nonparametric test between unpaired groups (Mann–Whitney test) of data was performed to establish the significance of results.

## 3. Results

### 3.1. Analytical Sensitivity of the Lumipulse^®^ G SARS-CoV-2 Ag Assay

The analytical sensitivity of the Lumipulse^®^ G SARS-CoV-2 Ag assay was evaluated by testing multiple replicates of a pool of saliva samples obtained from healthy donors spiked with serial dilutions of the INMI SARS-CoV-2 isolate, according to the scheme indicated in Table 2.

Results obtained from replicates of each aliquot were analyzed by insertion into a Probit regression curve (as shown in Figure 1A) to calculate the assay’s LOD, which resulted in 0.65 log_10_ TCID50/mL (confidence interval (CI), 0.44–1.58), i.e., 4.46 TCID50/mL, corresponding to 4.26 log_10_ copies/mL (CI, 4.04–5.11) or 18,197 copies/mL of SARS-CoV-2 RNA. As shown in Figure 1B, there was a linear correlation between SARS-CoV-2 antigens and RNA concentrations (r = 0.99; *p* < 0.0001).

### 3.2. Performance of the Lumipulse^®^ G SARS-CoV-2 Ag Assay with Frozen Saliva Samples

Initially, we carried out our performance study of Lumipulse^®^ G SARS-CoV-2 Ag assay on 169 clinical samples, kept frozen at −80 °C for a median of 17 days (min-max: 5–34), after being tested for the presence of SARS-CoV-2 RNA 67 saliva samples had a positive result and 102 a negative result with the Simplexa™ COVID-19 Direct assay. Retrospective analysis with the Lumipulse^®^ G SARS-CoV-2 Ag assay showed a sensitivity of 53.7% (36/67 positive sample with both assays) and a specificity of 97.1% (99/102 negative sample with both assays), as shown in Table 3.

Median Ct value (23.8 Ct; range: 16.4–45 Ct) for samples collected <7 days from symptoms onset was significantly lower (*p* = 0.0036) than the corresponding value for samples collected >7 days from symptoms onset (34.5 Ct; range: 11.3–45 Ct). In parallel, the median Ag concentration for samples collected <7 days from symptoms (1.9 pg/mL; range: 0.01–4782 pg/mL) was significantly higher (*p* = 0.0103) with respect to the corresponding value for samples collected >7 days from symptoms onset (0.07 pg/mL; range: 0.01–46.5 pg/mL), as expected. When stratifying samples into groups based on RNA Ct ranges, we noticed that, in each group, the percentage of positives did not distribute proportionally with the viral load (higher Ct value corresponds to lower viral load) (Table 4).

Furthermore, there was no correlation (r = 0.27; *p* = 0.211) between antigen concentrations (range: 0.77 to 4782 pg/mL or log_10_ −0.11 to 3.68 pg/mL) and Ct values (range: 18.2–33.5), thus suggesting the possibility that freezing could have affected the reliability of these results (Appendix A).

### 3.3. Performance of the Lumipulse^®^ G SARS-CoV-2 Ag Assay with Fresh Saliva Samples

Because of previous inconsistent results, we also tested 127 fresh saliva samples within 1 to 3 days of collection and kept them at +4 °C to evaluate the clinical performance of the assay. Among these samples, 42 were positive, and 85 were negative with the Simplexa™ COVID-19 Direct assay for the presence of SARS-CoV-2 RNA. Among those with a negative RT-PCR result, 45 samples were from healthy donors or patients not diagnosed with COVID-19 (and then considered as true negative), and the remaining 40 samples were from patients with a previous diagnosis of COVID-19 (and then considered as recovered from COVID-19). Median Ct values for samples collected <7 days from symptoms (27.0; range: 21.0–45.0) onset was significantly lower (*p* = 0.0031) than the corresponding value for samples collected >7 days from symptoms onset (45.0; range: 19.9–45). In parallel, median Ag concentration for samples collected <7 days from symptoms (1.1 pg/mL; range: 0.01–4358 pg/mL) was significantly higher (*p* = 0.0282) with respect to corresponding value for samples collected >7 days from symptoms onset (0.08 pg/mL; range: 0.01–699 pg/mL), as expected. As shown in Table 5, the overall agreement between Lumipulse and RT-PCR results was 80.3%, whereas the sensitivity and specificity were 52.4% and 94.1%, respectively.

However, excluding the patients recovered from COVID-19, the specificity of the antigenic assay increased to 100%. Unlike frozen samples, stratifying fresh samples into groups based on RNA Ct ranges (Table 6), the percentage of positives did distribute proportionally with the viral load, with greater antigen concentrations corresponding to higher viral loads (or lower Ct values).

Accordingly, the correlation between antigen concentrations (range, 0.77–4358 pg/mL or log_10_ −0.11 to 3.64 pg/mL) and Ct values (range, 18.3–30.1) was significant (r = 0.675; *p* = 0.0006) (Figure 2).

We anticipated predictive values of the Lumipulse^®^ G SARS-CoV-2 Ag assay based on the differing prevalence of SARS-CoV-2 infection (Table 7).

As expected, at low prevalence, the positive predictive value (PPV) was low, and the negative predictive value (NPV) was high (e.g., at 0.5% prevalence, the PPV was 4.28%, and the NPV was 99.75%). Notably, when considering the specificity with respect to the infection stage, the PPV was 100% at all prevalence values, whereas the NPV was unchanged. Considering that in a recent editorial, the low PPV of the Lumipulse SARS-CoV-2 Ag assay was pointed out as a possible severe test problem [15], and one possible solution was to raise the cut-off value of the assay, we simulated a progressive increase of the cut-off, until reaching 3.67 pg/mL. By setting this new cut-off, we observed a minor number of false-positive results (three instead of five), but five true positive results have been missed.

## 4. Discussion

New diagnostic methods, based on detection of viral antigens, could complement the COVID-19 diagnosis currently based on molecular testing [16], satisfying the pressing demand for an early diagnosis of SARS-CoV-2 infection due to their simple execution and short turnaround time (i.e., about 30 min) [3,4,5]. However, numerous recent studies aimed at evaluating the performance of rapid antigen tests highlighted their reduced sensitivity when used with clinical respiratory samples [4,5,6,17,18,19,20], and low sensitivity was also a critical issue reported when performed on saliva samples [9,21].

Here we described, for the first time, the good performance of Lumipulse^®^ G SARS-CoV-2 Ag assay for quantitatively measuring the SARS-CoV-2 N antigen on fresh saliva samples by chemiluminescent enzyme immunoassay technology. The low LOD of the assay, established using a pool of fresh saliva samples from healthy donors spiked with the INMI SARS-CoV-2 isolate, was 0.65 log_10_ TCID50/mL corresponding to 18,197 copies/mL of SARS-CoV-2 RNA. Moreover, a very high correlation between the viral N antigen and viral RNA concentrations was observed (r = 0.99; *p* < 0.0001). Then, we compared Lumipulse^®^ G SARS-CoV-2 Ag assay results on saliva samples collected from hospitalized patients with those obtained using Simplexa™ COVID-19 Direct assay. The first round of comparison, using frozen stored samples, provided unreliable results. The second round of comparison, using freshly collected samples, revealed an overall good agreement between positive/negative results (80.3%), showing a sensitivity of 52.4% and a specificity of 94.1%. Nevertheless, these values differ from those reported by the manufacturer (70.5% and 100%, respectively), this discrepancy is probably due to the different sample compositions (i.e., different percentage of subjects sampled in the early phases of the infection during which the viral load is high) and in the assay conditions (i.e., the time-lapse between the sample collection and assay execution). However, we observed increased assay specificity (from 94.1% to 100%) when PCR-negative patients with previous COVID-19 were removed from the analysis, probably due to a longer persistence of the antigen as compared to viral RNA in these samples, as observed for other infections, such as the Dengue virus [22].

One strength of this study is the demonstration of a significant correlation between quantitative antigen concentrations and RNA Ct values. It is to underline that a higher correlation was obtained when considering the analytical sensitivity (r = 0.99; *p* < 0.0001) as compared to the clinical sensitivity (r = −0.675; *p* = 0.0006). A possible explanation is that analytical sensitivity was performed on a homogeneous matrix spiked with serial dilutions of a single virus preparation, while the clinical sensitivity included samples from different subjects and, therefore, may suffer from individual differences in terms of viscosity or other factors, such as pH, the presence of spurious materials, etc.

Another strength of this study is to highlight that the result’s reliability of the Lumipulse^®^ G SARS-CoV-2 Ag assay strongly depends on the sample’s storage before testing. In fact, despite the manufacturer’s declaration of test suitability for frozen samples, we observed an evident difference in the performance of the assay using fresh or frozen saliva samples, which led us to consider stored saliva samples not practicable for diagnostic use with this assay. The suboptimal performance of the Lumipulse^®^ G SARS-CoV-2 Ag assay on frozen samples could be due to altered stability of viral protein(s) architecture by the freeze-thaw steps, which, in turn, could affect the recognition in the immunometric assay.

## 5. Conclusions 

We demonstrated that saliva could be a valid alternative to nasopharyngeal swab when a chemiluminescence-based assay, i.e., Lumipulse^®^ G SARS-CoV-2 Ag assay, is used for detection of SARS-CoV-2 antigen. In addition, we specify that the Lumipulse^®^ G SARS-CoV-2 Ag assay should be applied only to freshly collected saliva samples to reach a sensitivity greater than 90.0% for samples with RNA Ct values ≤25, a value considered as a good indicator of high viral load.

The strong dependence of antigen test results from the Ct value of the samples for freshly collected samples (Table 6), coupled with the significantly lower median Ct value for samples collected <7 days from symptoms onset, and the significantly higher median Ag concentration indicates that the best performances of the Lumipulse^®^ G SARS-CoV-2 Ag assay are obtained in patients at early stages of the infection.

It has to be considered that the performance of this automated antigen assay is deeply affected by the context of prevalence, as for antigen tests in general [23]. As stated above, antigen tests perform best when the person is tested in the early stages of infection with SARS-CoV-2, when viral load is generally highest. The low positive predictive value in the context of low prevalence for SARS-CoV-2 underscores the need for confirmatory molecular testing in SARS-CoV-2 antigen-positive cases. On the contrary, molecular confirmation is not necessary for settings with high pre-test probability, such as in symptomatic persons. On the other hand, to overcome the reduced sensitivity compared to molecular tests, repeated testing at weekly or bi-weekly frequency is recommended for monitoring purposes in asymptomatic low-risk populations, for surveillance purposes [2,24].

## Figures and Tables

**Figure 1 jcm-10-01471-f001:**
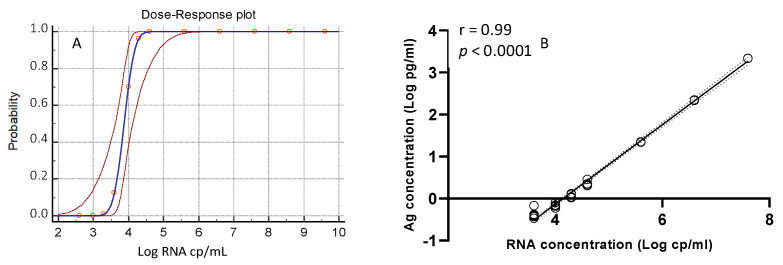
(**A**) Probit regression curve (blue line) with 95% of Confidence Interval (dashed red line). (**B**) Correlation between the antigen concentration (Log pg/mL) and SARS-CoV-2 RNA (Log copies/mL).

**Figure 2 jcm-10-01471-f002:**
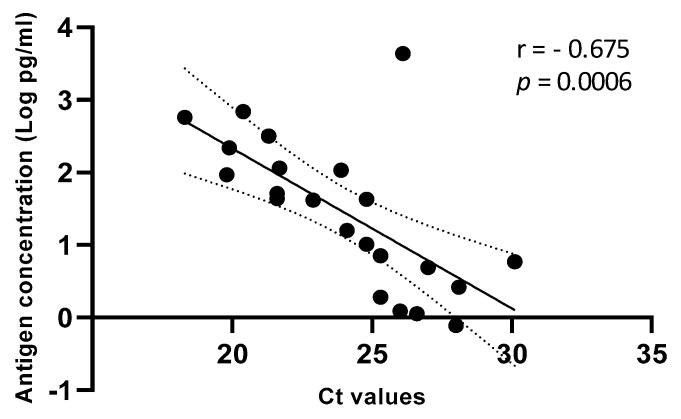
Correlation between Ag concentration (Log pg/mL) and Ct values on fresh saliva samples. Linear regression with 95% of Confidence Interval (dashed line).

**Table 1 jcm-10-01471-t001:** Description of saliva samples analyzed.

Samples Data	Patients’ Frozen Samples	Patients’ Fresh Samples	Healthy Donors’ Samples
Number of Donors	82	55	21
Number of Samples	169	87	40
Age (mean (min-max))	50 (19–87)	52 (20–89)	41 (25–54)
Gender (%F; (n. F/n. Donors))	25.6%; 21/82	43.6%; 24/55	76.2%; 16/21
Days from symptom onset (mean (min-max))	11 (0–37)	14 (1–43)	N.A.
Days ≤ 7 (*n*; %)	20; 46.5%	19; 24.7%	N.A.
Days > 7 (*n*; %)	23; 53.5%	58; 75.3%	N.A.

N.A. = not applicable; F = females.

**Table 2 jcm-10-01471-t002:** Analytical sensitivity (LOD) was determined with spiked SARS-CoV-2 on a saliva matrix from healthy donors by Probit analysis.

Viral Preparation (TCID_50_/mL)	RNA cp/mL *	Lumipulse G SARS-CoV-2 Ag
		Overall % Determinations (Replicates)
1,000,000	4 × 10^9^	100% (3/3)
100,000	4 × 10^8^	100% (3/3)
10,000	4 × 10^7^	100% (3/3)
1000	4 × 10^6^	100% (3/3)
100	4 × 10^5^	100% (3/3)
10	4 × 10^4^	100% (5/5)
5	2 × 10^4^	100% (5/5)
2.5	1 × 10^4^	60% (3/5)
1	4 × 10^3^	16.67% (1/6)
0.1	4 × 10^2^	0% (0/3)
Probit analysis		
LOD: TCID_50_/mL		4.46
LOD: RNA cp/mL		18,197

* RNA copies/mL were calculated on standard curve for gene E SARS-CoV-2.

**Table 3 jcm-10-01471-t003:** Comparison of Lumipulse G SARS-CoV-2 Ag data vs. molecular reference test (Simplexa ™ COVID-19 Direct) on frozen retrospective saliva samples.

Comparison of Frozen Saliva Samples		Lumipulse G SARS-CoV-2 Ag		
		Positive	Negative	Total
**Simplexa™ COVID-19 Direct**	**Positive**	36	31	67
	**Negative**	3	99	102
	**Total**	39	130	169
		**Proportion #**	**Percentage (95% CI)**	
**Sensitivity**		36/67	53.7% (41.12–66.0%)	
**Specificity vs. RT-PCR** **reference test**		99/102	97.1% (96–99.4%)	

# n. pos./N. Tot.

**Table 4 jcm-10-01471-t004:** Percentage of positivity of frozen saliva samples according to the Ct range of the molecular test.

Ct Ranges	Ag Positive Samples/Total Positive PCR	Positivity % with Lumipulse G SARS-CoV-2 Ag
<20	4/8	50.0%
20–25	11/15	73.3%
25.01–30	13/16	81.3%
>30	8/28	28.6%

**Table 5 jcm-10-01471-t005:** Performance of Lumipulse G SARS-CoV-2 Ag vs. reference molecular test (Simplexa™ COVID-19 Direct) on fresh saliva samples.

Comparison of Fresh Saliva Samples		Lumipulse G SARS-CoV-2 Ag		
		Positive	Negative	Total
**Simplexa™ COVID-19 Direct**	**Positive**	22	20	42
	**Negative**	5	80	85
	**Total**	27	100	127
		**Proportion #**	**Percentage (95% CI)**	
**Sensitivity**		22/42	52.4% (36.4–68.0%)	
**Specificity vs. RT-PCR** **reference test**		80/85	94.1% (86.8–98.1%)	
**Specificity vs. stage of infection**		45/45	100% (92.1–100.0%)	

# n. pos./N. Tot.

**Table 6 jcm-10-01471-t006:** Percentage of positivity of fresh saliva samples based on Ct ranges.

Ct Range	N° of Ag Positive Samples/Total Positive PCR	Positivity % with Lumipulse G SARS-CoV-2 Ag
<20	3/3	100%
20–25	10/11	90.9%
25.01–30	8/14	57.1%
>30	1/14	7.1%

**Table 7 jcm-10-01471-t007:** Simulation of Positive Predictive Value, Negative Predictive Value and accuracy based on prevalence.

Value 95% of CI	Prevalence of Infection	Positive Predictive Value *	Negative Predictive Value *	Accuracy *
Value	0.5%	4.28%	99.75%	93.91%
95% CI		1.79–9.90%	99.65–99.82%	88.23–97.38%
Value	1%	8.25%	99.49%	93.70%
95% CI		3.54–18.08%	99.30–99.63%	87.96–97.24%
Value	2%	15.38%	98.98%	93.28%
95% CI		6.89–30.85%	98.60–99.26%	87.44–96.96%
Value	10%	49.73%	94.68%	89.94%
95% CI		28.73–70.83%	92.80–96.08%	83.35–94.57%

* These values depend on the prevalence.

## Data Availability

The data presented in this study are available on request from the corresponding author.

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
