# Peer review of "Saliva Is a Valid Alternative to Nasopharyngeal Swab in Chemiluminescence-Based Assay for Detection of SARS-CoV-2 Antigen"

_jcm, 2021, doi:10.3390/jcm10071471_

Round 1

Reviewer 1 Report

This paper is the test result of SARS-CoV-2 antigen in Saliva sample.

The treatise is well organized and useful.

Minor 

Table 1 Supplementary explanation such as F in the table is required.

Author Response

  1. Table 1 Supplementary explanation such as F in the table is required.

Answer: F is for female gender. It is now specified in the table

Reviewer 2 Report

Thank you for submitting the manuscript to Journal Clinical Medicine!

I have the following comments, and look forward to the revised manuscript.

  1. Please include the verified LOD of the Simplexa SARS-CoV-2 RT-PCR assay (paragraph 2.4); Also, please clarify "an arbitrary value of 45 was assigned to negative (Ct >40) samples" and its relevance in this study.
  2. It was mentioned that the infectious virus stock concentration was 107 TCID15/mL (or 1010 RNA copies/mL). But it is not clear to me what's the viral concentration in the propagated preparation from cultured VERO E6 cells (paragraph 2.2), used for the LOD verification. I did note that in Table 2, the start concentration was 1,000,000 TCID50/mL (4x10e9 copies/mL). 
  3. Table 1: Were the saliva samples (20+23=43 frozen saliva samples; 19+58=77 fresh saliva samples) collected from confirmed COVID-19 patients? If Yes, the Ct vales and antigen test results (<7 days versus >7 days from onset of symptoms) should be presented in Results and discussed in Discussion sections, respectively.
  4. Please clarify pg/mL: Is it concentration of the N antigen in saliva sample?
  5. In Discussion section, please discuss the use and algorithm of antigen test in symptomatic (diagnostic testing) versus asymptomatic (monitoring testing) settings, and the recommended frequency of testing for monitoring (surveillance).
  6. Please discuss the likely effects of freeze/thaw of saliva samples on the antigen test (? stability of N antigen in thawed frozen sample).
  7. Lines 39-40: please update the number of COVID-19 cases and confirmed deaths.
  8. Please correct typo errors such as "freshly" (line 23), "No" (Table 4), "25,01" (Table 4), etc. Please check the manuscript thoroughly for grammatical and typo errors.

Author Response

I have the following comments, and look forward to the revised manuscript.

  1. Please include the verified LOD of the Simplexa SARS-CoV-2 RT-PCR assay (paragraph 2.4); Also, please clarify "an arbitrary value of 45 was assigned to negative (Ct >40) samples" and its relevance in this study.

Answer: The LOD has been added (page 4, lines 127,129).

Considering the second point (an arbitrary value of 45 was assigned to negative (Ct >40) samples) a sentence was added to clarify (page 4, lines 142-143, 152-154).

  1. It was mentioned that the infectious virus stock concentration was 107 TCID15/mL (or 1010 RNA copies/mL). But it is not clear to me what's the viral concentration in the propagated preparation from cultured VERO E6 cells (paragraph 2.2), used for the LOD verification. I did note that in Table 2, the start concentration was 1,000,000 TCID50/mL (4x10e9 copies/mL). 

Answer: Thank you for notifying the possible confusing point. It has been clarified (page3, lines 93-101, 105-107)

  1. Table 1: Were the saliva samples (20+23=43 frozen saliva samples; 19+58=77 fresh saliva samples) collected from confirmed COVID-19 patients? If Yes, the Ct vales and antigen test results (<7 days versus >7 days from onset of symptoms) should be presented in Results and discussed in Discussion sections, respectively.

Answer: Both frozen and fresh saliva samples derived from confirmed COVID-19 patients.

As requested, median of Ct values and Ag concentrations have been added in the Results (page 6, line189-195; 213-219) and discussed in the appropriate section (Discussion, page 9, lines 300-304).

  1. Please clarify pg/mL: Is it concentration of the N antigen in saliva sample?

Answer: It is N antigen concentration, indeed. The information has been added

  1. In Discussion section, please discuss the use and algorithm of antigen test in symptomatic (diagnostic testing) versus asymptomatic (monitoring testing) settings, and the recommended frequency of testing for monitoring (surveillance).

Answer: according to reviewer suggestion, a sentence concerning the use and algorithm of antigen test has been added (page 10, Lines 305-314)

  1. Please discuss the likely effects of freeze/thaw of saliva samples on the antigen test (? stability of N antigen in thawed frozen sample).

Answer: a sentence discussing likely effects of freeze/thaw of saliva samples on the antigen test has been added (page 9, Lines 291-293)

  1. Lines 39-40: please update the number of COVID-19 cases and confirmed deaths.

Answer: The numbers were updated and also the corresponding reference

  1. Please correct typo errors such as "freshly" (line 23), "No" (Table 4), "25,01" (Table 4), etc. Please check the manuscript thoroughly for grammatical and typo errors.

Answer: Done

Reviewer 3 Report

Amendola et al, presented a very interesting article with regards of new alternatives for the detection of SARS-COV2 on saliva, using a chemiluminescence-based assay. The methodology is robust and reflects the high quality of the investigators work. The reviewer has only minor comments that the reader might find useful to have. Overall, it would be useful for the reader to have the investigators opinior regarding the advantages of this new assays on the laboratory daily practice and the clinical impact of introducing this assay as a routine technique, such as:

What is the volumen of saliva that would be needed to perform the chemiluminescence test?

What is the time on hands to perform the assay?

How many samples can be performed simultaneously?

Kind regards

Author Response

Overall, it would be useful for the reader to have the investigators opinion regarding the advantages of this new assays on the laboratory daily practice and the clinical impact of introducing this assay as a routine technique, such as:

  1. What is the volumen of saliva that would be needed to perform the chemiluminescence test?

Answer: The volume of saliva was specified (Page 3, lines 115-116)

  1. What is the time on hands to perform the assay?

Answer: This information was added (Page 3, Line 118)

  1. How many samples can be performed simultaneously?

Answer: This information was added (Page 3, Line 118)